# Is There a Relationship between Anthropometric Indices and Muscular, Motor, and Cardiorespiratory Components of Health-Related Fitness in Active European Older Adults?

**DOI:** 10.3390/ijerph21020201

**Published:** 2024-02-09

**Authors:** José Mª Cancela-Carral, Elena Vila, Iris Machado, Gustavo Rodríguez, Adriana López, Bruno Silva, Pedro Bezerra

**Affiliations:** 1Faculty of Education and Sports Science, University of Vigo, 36005 Pontevedra, Spain; evila@uvigo.gal (E.V.); adrianalpez102@gmail.com (A.L.); 2HealthyFit Research Group, Galicia Sur Health Research Institute (IIS Galicia Sur), Sergas-UVIGO, 36213 Vigo, Spain; irismacoli@uvigo.gal (I.M.); gfuentes@uvigo.gal (G.R.); 3Faculty of Physiotherapy, University of Vigo, 36005 Pontevedra, Spain; 4Polytechnical Institute of Viana do Castelo, 4900-347 Viana do Castelo, Portugal; silvabruno@esdl.ipvc.pt (B.S.); pbezerra@esdl.ipvc.pt (P.B.); 5Research Center in Sports Performance, Recreation, Innovation and Technology—SPRINT, 4960-320 Melgaço, Portugal

**Keywords:** aging, obesity, health, quality of life, physical activity

## Abstract

The aging process induces alterations in the body, resulting in changes in both health-related fitness and specific anthropometric measures. These changes often pose health risks for older adults. The aim of the present study was to investigate whether there is an association between anthropometric indices and muscular, motor, and cardiorespiratory components of health-related fitness in active European older adults. This study included 2687 European older adults, comprising 1999 women and 688 men, with an average age of 70.05 ± 5.5 years. The assessment included health-related fitness using the Senior Fitness Test and anthropometric indices, such as the body adiposity index, body mass index, conicity index, waist-to-hip ratio, and waist-to-height ratio, among others. The results indicated that gender significantly influences the values of physical performance and anthropometric parameters, making them incomparable. The degree of correlation between anthropometric indices and muscular, motor, and cardiorespiratory components of fitness depends on each anthropometric index analysed. The anthropometric index most correlated with physical fitness performance parameters is the waist-to-height ratio (WHR), followed by the body mass index (BMI). Cardiorespiratory endurance and balance are the two physical parameters most correlated with anthropometric indices.

## 1. Introduction

Population aging is now one of the greatest challenges for public health. It is an intrinsic process of demographic transition and a shift in the age structure of the population, considered one of the most significant social transformations of the 21st century and one of the greatest achievements that humanity has attained in contemporary times. Since the early 21st century, the world has been aging rapidly due to a significant decline in fertility rates, coupled with an increase in life expectancy. The year 2050 will be a demographic turning point, as it is expected that 22% of the world’s population will be 60 years and older, representing over two billion people, compared to 12% in 2015 (900 million) [1].

There are numerous definitions of aging, all of which identify it as a complex, dynamic, and multifactorial process inherent to every human being, involving a set of morphological and physiological changes that occur as a result of the passage of time [2,3]. From a biological perspective, aging is a process of accumulating molecular and cellular damage in a complex interaction between damage, maintenance, and repair that gradually reduces physiological reserves, increases the risk of diseases, and generally diminishes the physical and mental capacity of the individual, leading to an increasing risk of disease and disability [4]. This aging process causes morphological and physiological modifications in the body, such as an increase in body weight, a decrease in musculoskeletal mass, loss of strength, and a reduction in the contractibility of muscle fibres [5,6,7], negatively impacting functional capacity and the perception of quality of life in this population [8]. All the changes indicated above are accentuated by the loss and deterioration of health-related fitness [9]. These alterations are influenced by the environment, the individual’s behaviours and healthy habits such as physical exercise and proper nutrition, but also by unhealthy habits such as smoking, alcohol consumption, and a sedentary lifestyle, among others.

Health-related fitness is, then, considered a state characterized by the ability to perform daily activities with vigour, and includes traits and abilities that are associated with a low risk of premature development of hypokinetic diseases [10,11]. Bouchard, Shepard, and Stephens [12] describe health-related fitness as the state of physical and physiological characteristics that define the risk levels of the premature development of diseases or morbid conditions that are associated with a sedentary lifestyle. A person’s health-related physical fitness can be expressed in five main components: 1. morphological component; 2. muscular component; 3. motor component; 4. cardiorespiratory component; and 5. Metabolic component [12].

Currently, the majority of individuals reaching old age exhibit reduced health-related fitness due to the presence of physical and cognitive problems [13]. Having high levels of health-related fitness is associated with a lower risk of chronic diseases in old age [14,15]. Therefore, it is necessary to promote healthy and active aging, understood as a process of optimising opportunities for health, participation, and safety to ensure the quality of life of older adults as they age. The goal of active aging is to promote and maintain health, well-being, and quality of life by helping to reduce stress, improve cognition, enhance thinking abilities, and strengthen functional and physical skills [16].

Various studies have indicated that motor, muscular, and cardiorespiratory components and certain anthropometric indices are related to each other and can be used as health indicators in individuals, particularly in older adults [17,18]. On the other hand, Gomez [19] concluded that there is a direct relationship between BMI and balance, indicating that a higher BMI is associated with poorer body stability. Similar results were found by Valdés-Badilla [20], who noted that excess body weight can limit upper body flexibility, agility, and balance in physically active older males. Other anthropometric indices, such as waist-to-hip ratio (WHR), waist-to-height ratio (WHtR), body adiposity index (BAI), and conicity index (CI), are also commonly used clinically as indicators of health status [21]. Currently, there is little evidence that relates these indices to the motor, muscular, and cardiorespiratory components that define health-related fitness [22]. These findings highlight that nutritional status together with functional status (physical and mental) can interfere with the ability of older adults to carry out activities of daily living independently, autonomously, and satisfactorily. Factors such as the level of muscle strength and endurance, the maintenance of joint flexibility, motor skills, and comorbidities clearly influence the functional capacity of older adults [23]. Therefore, anthropometric and age-related body composition changes are increasingly being taken into account in clinical practice due to their importance to and relationship with the health of older adults. Changes in weight, height, and body composition are identified in older adults, as it has been established that in old age, there is an increase in skinfold thickness, waist-to-hip ratio, and body mass index (BMI), among other factors [24]. All these changes make older adults more vulnerable, which is further accentuated by potential complications of various diseases, compromising their health status and functional capacity [25].

Based on the aforementioned factors, the present research study was proposed with the objective of determining the following: is there an association between anthropometric indices and muscular, motor, and cardiorespiratory components of health-related fitness in active European older adults?

## 2. Materials and Methods

### 2.1. Participants

The recruitment of participants was carried out in compliance with the Declaration of Helsinki and was approved by the Ethics Committee of the Polytechnic Institute of Viana do Castelo (IPVC-ESDL180417). Informed consent was obtained from all participants. Inclusion criteria were as follows: age between 59 and 90 years and participation in the In Common Sport Plus Project in the years 2021–2023. Exclusion criteria included any functional or pathological limitations that would impede the normal development of any of the proposed tests. In this study, 2687 active European older adults participated (Spain, Italy, Slovenia, Bulgaria, Hungary, and Portugal), 1999 of whom were women and 688 of whom were men, with an average age of 70.05 ± 5.5 years (Figure 1). We defined active older adults as all those who met the recommendations of the American College of Sports Medicine regarding physical exercise [26,27]. Data were collected and analysed in a way that prevents the direct identification of participants, due to data coding.

### 2.2. Evaluation

The examiners were experts in the use of anthropometric indices and the evaluation of body composition parameters through bioelectrical impedance analysis (BIA). During the measurements, participants were barefoot, wore minimal clothing, and conditions were strictly standardized as established by the manufacturer and previous research. 

Bioelectrical impedance analysis (BIA) was performed using a multi-frequency segmented body composition analyser (TANITA MC-780 S MA, Tanita Corporation, Tokyo, Japan). The analysis of all components of body composition took approximately 20 s, and body mass was also measured with an accuracy of 0.1 kg. The analyser took into account age, gender, and height without a decimal point [28,29]. Height was measured with the subject standing, barefoot, using a portable stadiometer (SECA, 213) with an accuracy of 1 mm. Weight was measured with the subject standing, without shoes or socks, using TANITA MC-780 S MA with an accuracy of 0.1 kg. Body mass index (BMI) was calculated using (TANITA MC-780 S MA, Tanita Corporation, Tokyo, Japan), to which the participant’s height, previously measured, was incorporated. Waist circumference (WC) was measured in an area between the lower rib and the iliac crest, in upper third. Hip circumference (HC) was measured in the most prominent point of the hip. Measurements of WC and HC were measured with a non-stretchable tape measure, with the scale in centimetres, and were performed twice. If the difference was less than 1 cm, the mean of the two measures was recorded and if the difference was more than 1 cm, both measurements were repeated [30,31].

Waist-to hip ratio (WHR) was calculated as WC (cm)/HC (cm) and waist-to-height ratio (WHtR) as WC (cm)/height (cm) [30]. Body adiposity index (BAI) is a non-linear ratio of hip circumference (HC) to height. BAI was calculated using the following formula: BAI = HC in cm/{[height in m]1.5–18}) [32]. CI was calculated from the relationship of waist circumference with height and weight using the formula CI = WC in m/[0.109 × √ {Body weight in kg/Height in m}]), where 0.109 is a constant resulting from the conversion of volume and mass units to length units [33].

To assess physical condition, the Senior Fitness Test (SFT) protocol was followed, as previously described and validated for self-sufficient individuals without health problems aged 60 to 94 years [34]. The tests employed based on the analysed physical parameter were Arm Curl and Chair Stand Test (strength), Chair Sit and Reach and Back Scratch (flexibility), Eight-Foot Up-and-Go (balance), and Six-Minute Walk and Two-Minute Step Test (cardiorespiratory endurance). 

Muscle strength was evaluated through three tests:Arm Curl: The participant is required to perform the maximum number of elbow flexion–extension repetitions with a dumbbell (2 kg for women and 4 kg for men) in 30 s. The test is performed with the dominant arm. The participant starts seated in a chair with a straight back and feet flat on the floor. They grasp the dumbbell with the dominant hand, placing it perpendicular to the ground, with the palm facing the body and the arm extended. They lift the weight from this position, gradually rotating the wrist (supination) until completing the arm flexion movement, with the palm facing upward. They return the arm to the starting position by performing a full extension of the arm, now rotating the wrist toward the body. At the “go” signal, the participant performs this complete movement as many times as possible within 30 s. At first, the exercise is demonstrated slowly for the participant to observe the correct execution, and then the speed is increased to illustrate the rhythm. For the correct execution, only the forearm should move, keeping the arm steady (keeping the elbow close to the body can help maintain this position);Chair Stand Test: The participant starts sitting in the middle of the chair with his or her back straight, feet on the floor, and arms crossed over the chest. From this position and at the “go” signal, the participant should stand up completely and return to the initial position as many times as possible within 30 s. It is recommended that the exercise be demonstrated slowly first so that the participant can observe its correct execution, followed by a faster demonstration for better comprehension. Before starting the test, the participant shall perform the exercise one or twice to ensure the correct execution.

In addition to tests related to the Senior Fitness Test battery, muscle strength was assessed using a handgrip strength technique using isometric hand dynamometry:3.Grip Strength Test: The reason for incorporating this test was because it is widely used in older adults as an indicator parameter of frailty syndrome, which is closely linked to health and changes in body weight among older adults [35]. Grip strength is a simple and suitable measure to assess the muscle strength levels of older adults. Isometric dynamometry involves measuring the force or tension exerted against static resistance. Mechanical dynamometers designed to measure a single muscle group are used for this purpose. In this study, the manual dynamometry test was performed using a Jamar^®^ Hydraulic dynamometer (J.A. Preston Corporation, Clifton, NJ, USA), recording force in kilograms, under the following conditions: Prior adjustment of the dynamometer grip was made according to the size of the hand. The grip was adjusted so that the subject’s proximal interphalangeal joint, when gripping the dynamometer, formed a 90° angle. The subject was seated with arms bent at 90° and supported on a table. They were instructed to exert as much force as possible without lifting the arm off the table. Three attempts were made, and the highest value was recorded. The test was performed on the dominant arm. Manual muscle strength equal to or greater than 30 kg for men and equal to or greater than 20 kg for women was considered normal; values below these were deemed inadequate [36,37].

Flexibility was evaluated through two tests:Chair Sit and Reach: The participant sits on the edge of the chair with one leg flexed and the other fully extended. They perform trunk flexion without bouncing, attempting to touch the toes of the extended leg. Values are measured and recorded in centimetres; a positive result is achieved when the participant reaches beyond the tips of their toes, and a negative result when they do not reach them. The participant sits on the edge of the chair (the fold between the upper leg and the buttocks should rest on the front edge of the seat). One leg is bent with the foot on the floor, while the other leg is extended as straight as possible in front of the hip. With arms outstretched, hands together, and middle fingers aligned, the participant flexes at the hip slowly, attempting to reach their toes or surpass them. If the extended leg starts to flex, the participant returns to the starting position until the leg is fully extended. The participant must hold this position for at least 2 s. The participant performs the test on each leg to determine the better-performing leg (the final test is performed with the better of the two). Before starting the test, the participant will perform a brief warm-up with a couple of attempts using the preferred leg;Back Scratch: The participant should touch their fingertips behind their back. The dominant arm passes over the shoulder, and the other under the shoulder. Values are measured and recorded in centimetres; a positive result is achieved when the fingertips touch each other, and a negative result when the fingertips do not touch. The participant stands with the preferred arm above the shoulder, palm down, and fingers extended. From this position, they bring the hand towards the middle of the back as far as possible, keeping the elbow up. The other arm is placed on the back, behind the waist, with the palm up and reaching as far as possible, attempting to touch the middle fingers of both hands. The participant should practice the test to determine the best side and may perform it twice before starting the actual test. They should ensure that the middle fingers of one hand are as closely aligned as possible with the middle fingers of the other hand; the examiner may guide the participant’s fingers (without moving their hands) to achieve proper alignment. Participants are not allowed to grasp and pull on their fingers.

Balance was evaluated through a single test:Eight-Foot Up-and-Go: Starting from a seated position, the participant must stand up, walk 2.44 m, pass behind a cone, and sit down again as quickly as possible. The execution time of this action, performed at a fast walking pace, is measured. The participant will sit in the middle of the chair with his or her back straight, feet on the floor, and hands on their thighs. One foot will be slightly ahead of the other, and the trunk slightly bent forward. At the “go” signal, the participant will stand up and walk as fast as possible around the cone and sit down again. Time is counted from the moment we say “go”, even if the participant has not started moving. The time will stop when the participant is seated in the chair.

Cardiorespiratory endurance: aerobic resistance was evaluated through two tests:Two-Minute Step Test: The subject must perform alternate knee lifts to the point marked on the wall (mid-thigh level) as many times as possible (repetitions) in two minutes. One repetition is counted once the right and then the left leg have been lifted. At the “go” signal, the participant begins to march in place as many times as possible for 2 min. If the participant does not reach this benchmark, they will be asked to slow down the pace to ensure the validity of the test without stopping the time.Six-Minute Walk: This is a cardiorespiratory functional test that measures the maximum distance a subject can walk in 6 min. Widely used to assess the progression and quality of life of patients with cardiorespiratory diseases, it is considered an easy-to-administer, well-tolerated test that reflects the activities of daily living. This test will be conducted once all previously performed tests are completed. Each participant will start every 10 s. At the “go” signal, the participant will walk as fast as possible for 6 min following the marked circuit. To count the number of laps completed, the examiner will give a stick to the participant for each lap or mark it on the record sheet. At the 3 and 2 min marks, participants will be notified of the remaining time to regulate the pace of the test. After 6 min, the participant will move to the right and stand on the nearest mark while slowly raising their legs alternately to keep moving.

### 2.3. Statistical Analysis

All quantitative variables were described using the mean and standard deviation, and qualitative variables were presented as percentages. The normal distribution of continuous variables was tested using the Kolmogorov–Smirnov test (*p* > 0.05). The existence of significant differences (male vs. female) was analysed through the independent samples *t*-test (quantitative variables) and chi-square test (qualitative variables). The association between anthropometric indices and muscular, motor, and cardiorespiratory components of health-related fitness was visualized through the lowess fit plot. The z-scores of BMI, WHR, WHtR, BAI, and CI were used to standardize the anthropometric indices for comparability. The association between the anthropometric indices and muscular, motor, and cardiorespiratory components was analysed through linear regression with adjustment for age and gender since age and gender can influence levels of health-related fitness. Each linear regression analysis included the anthropometric index under study and the different variables (muscular, motor, and cardiorespiratory components). To conclude the statistical analysis, analysis of variance with the Tukey-B post hoc test was performed to assess the behaviour of each physical parameter based on the obesity levels determined by BMI, WHR, WHtR, BAI, and CI. All analyses were conducted using IBM-SPSS version 29 software (IBM Inc., Armonk, NY, USA). The significance level established for the tests was *p* < 0.05, with a confidence level of 95%.

## 3. Results

Table 1 presents the characteristics of the sample, both overall and stratified by gender. A total of 2687 participants were analysed, including 688 men and 1999 women. The mean age of the sample was 70.05 ± 5.51. The highest percentage of participants was in the 65–69 age group (n = 1008, 37.50%). Muscular, motor, and cardiorespiratory components of health-related fitness were assessed by tests that measure levels of muscle strength, upper and lower body flexibility, aerobic endurance, and balance. Body composition was assessed using the following anthropometric parameters: height (cm), weight (kg), BMI (kg/m^2^), fat%, WC (cm), HC (cm), WHR, WHtR, BAI (%), and CI. The results indicated significant differences by gender in all the parameters evaluated, except for WHtR (men = 0.58 ± 0.06 vs. women = 0.58 ± 0.08). Men presented better values in all parameters except flexibility, where the pattern was reversed (Chair Sit and Reach: male = −1.55 ± 9.70, female = 1.56 ± 8.11; Back Scratch: male = −12.07 ± 15.21, female = −4.91 ± 11.90). The highest and most significant values were observed in WC, WHR, and CI for men, and in BMI, fat%, HC, and BAI for women.

Figure 2 visualises the association of physical fitness performance with anthropometric indices according to the z-score of BMI, WHR, WHtR, BAI, and CI using Lowess fit plots. In these plots, we can observe how strength levels tend to decrease as WHR, WHtR, and CI values increase. However, the opposite occurs when analysing BMI. If we examine cardiorespiratory endurance and flexibility, we can note that they exhibit a similar behaviour, as their values increase simultaneously with BMI, WHtR, and BAI, and tend to decrease when WHR and CI values increase. Notably, flexibility shows a drastic decrease with an increase in WHR. Finally, regarding dynamic balance, we observe that its behaviour is different from the other variables, as it remains relatively constant with small oscillations with higher BMI and WHR, increases with higher CI, and decreases with higher WHtR and BAI.

Table 2 presents the linear regression analysis, controlled for age and gender, of anthropometric indices in relation to muscular, motor, and cardiorespiratory components of health-related fitness. The results show that only WHtR has correlations with the four studied parameters of physical fitness performance, with cardiorespiratory endurance having the highest correlation (r = −0.242; *p* < 0.001). BMI, BAI, and CI show correlations with three out of the four parameters, while WHR only correlates with two (flexibility and cardiorespiratory endurance). Considering correlation behaviour, it is noteworthy that cardiorespiratory endurance exhibits an inversely proportional correlation (−0.266 to −0.080) with all indices, while balance shows a direct correlation (0.042 to 0.272).

In Table 3, the results of physical fitness performance parameters are shown based on different cutoff points of BMI, WHR, WHtR, BAI, and CI established as indicators of obesity levels. If we consider the BMI cutoff point, we can observe that its behaviour is influenced by the specific physical variable analysed. Those with higher BMI (>35) tend to have less muscular strength, while those with lower BMI (<27) are more flexible, have better balance, and demonstrate improved cardiorespiratory endurance.

Focusing on WHR analysis, we can observe that active older adults with higher WHR values (>0.90 for males; >0.85 for females) tend to be stronger and have better balance. On the contrary, those with lower WHR values (<0.90 for males; <0.85 for females) are more flexible and exhibit better cardiorespiratory endurance. The analysis of WHtR, BAI, and CI showed a similar tendency to that observed for BMI concerning the different physical parameters analysed.

## 4. Discussion

Aging leads to a decline in physical capabilities, resulting in a deterioration of physical fitness and a reduction in functionality [41]. Identifying associations between anthropometric indices and parameters of muscular, motor, and cardiorespiratory components of health-related fitness will help detect potential health alterations more quickly and accurately, pinpointing crucial aspects of health and enabling effective and correct interventions. In this study, 2687 older adults were analysed, and the results showed that WHtR is the anthropometric index with the strongest association with physical fitness performance parameters, with cardiorespiratory endurance being the parameter with the highest degree of correlation. However, this correlation is conditioned by WHtR levels, as high and low WHtR values show greater correlation than medium values (−2.5–2.5). These results align with those obtained by Chih-Hui-Chiu [42], who observed, after implementing aerobic programs of different intensities in young adults, that improvements in cardiorespiratory endurance were also associated with improvements in WHtR, with these improvements being conditioned by the training program implemented. Another research study on the relationship between WHtR and cardiorespiratory endurance was conducted by Mondal and Mishra [43], who concluded that high levels of Vo2max are related to low WHtR values, which contradicts the findings presented by Chih-Hui Chiu [42] and our study. Hence, further research is necessary to continue exploring the association between WHtR and cardiorespiratory endurance. The relationship between cardiorespiratory endurance and WHtR may be associated with low levels of fat mass and its distribution, since the concentration of fat mass in the abdominal region in older adults is common, and this concentration can be reduced by aerobic physical exercise.

BMI, BAI, and CI showed correlations with three of the four defined physical parameters, while WHR only correlated with two physical parameters (flexibility and cardiorespiratory endurance). High BMI levels (>35) were associated with low muscle levels, while low BMI levels (<27) were related to high values of balance, flexibility, and cardiorespiratory endurance. These results align with a study by Chen et al. [44], where optimal BMI (19–24 kg/m^2^) was associated with better physical performance, while BMI (>24) was associated with decreased physical performance. Faramarzi, Bagheri, and Banitalebi [45] conducted a study to determine the effect of a combined strength and aerobic programme on anthropometric indices, observing that improvements in aerobic endurance (Vo_2_max) were related to a decrease in BAI, results that are consistent with those obtained in this study. The effect of sequence order of combined strength and endurance training on new adiposity indices in overweight older adult women was studied by Ryu [46], who conducted a study to analyse the association between conicity index (CI) and cardiovascular risk factors (handgrip strength), observing that those with high CI values had low strength levels; these results are consistent with those obtained in this research study. The correlations identified between BMI, BAI, IQ, and physical parameters (flexibility, balance, cardiorespiratory endurance) reveal the influence of weight and fat mass distribution in older adults as conditioning factors of these correlations.

On the other hand, it was found that high WHR levels (>0.90 male; >0.85 female) are associated with high values of strength and balance, while low WHR values are associated with high values of flexibility and cardiorespiratory endurance. Lockie [47] also found positive correlations between WHR and strength levels (push-ups), indicating that this correlation varies depending on the muscle group analysed. Another study conducted by Lockie [48] reflected that WHR has an inverse relationship with flexibility (sit and reach) and aerobic endurance, results that confirm those presented in our research.

The results obtained in this study reveal the relationship between anthropometric indices and the muscular, motor, and cardiorespiratory components of health-related fitness, with correlations of different degrees depending on the weight of older adults and the distribution of fat mass in their bodies. Based on our results, we recommend the use of the WHtR and BMI anthropometric indices in older adults, since alterations in these may be related to risk factors linked to cardiovascular pathologies.

There are limitations to this study that should be pointed out. This study is characterized by its cross-sectional design, so it would be advisable to carry out a longitudinal follow-up study with the aim of ratifying the associations between anthropometric indices and motor, muscular, and cardiorespiratory parameters found here. Another limitation that we would like to highlight is the small percentage of older adults between 80 and 90 years old who have participated, which limits the applicability of our conclusions for this age group. We would also like to highlight the scarcity of research using BAI and CI, which complicates comparisons of results. On the other hand, we emphasize the strengths of the presented work, which include its large sample size (2687) and the analysis of the relationship between anthropometric indices and motor, muscular and cardiorespiratory parameters for the first time for this group, with graphs that show their correlations in a population of active older adults (see Figure 2).

## 5. Conclusions

This study made it possible to identify the existence of associations of varying degrees between anthropometric indices and muscular, motor, and cardiorespiratory components of health-related fitness in active European older adults. Data analysis identified waist-to-height ratio (WHtR) and body mass index (BMI) as the indices most strongly correlated with healthy physical performance. Cardiorespiratory endurance and balance are the parameters that show the highest degree of association with anthropometric indices. The analysis of health-related fitness based on obesity cutoff points (anthropometric indices) made it possible to establish health-related physical fitness levels for each of them.

## Figures and Tables

**Figure 1 ijerph-21-00201-f001:**
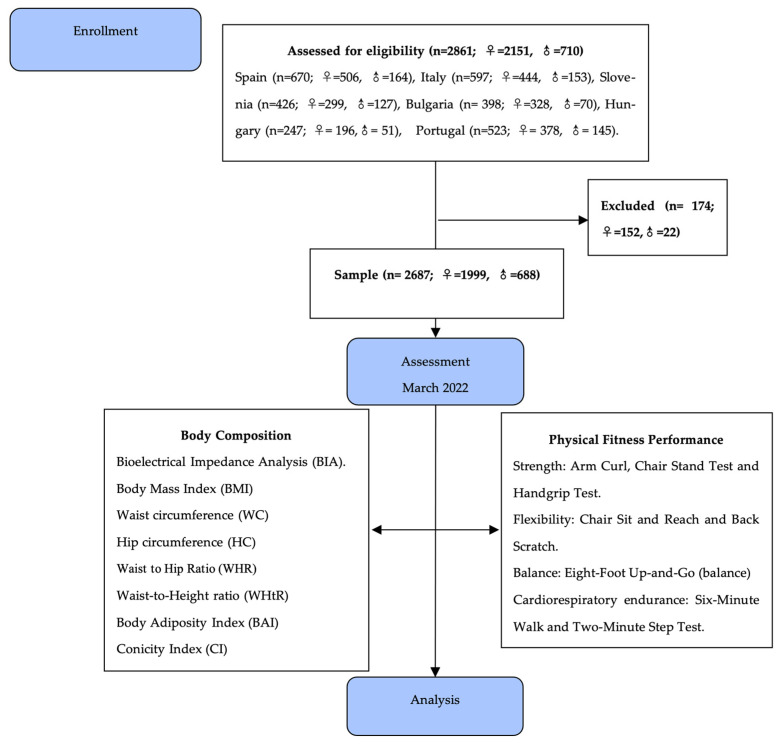
Flowchart of the study design.

**Figure 2 ijerph-21-00201-f002:**
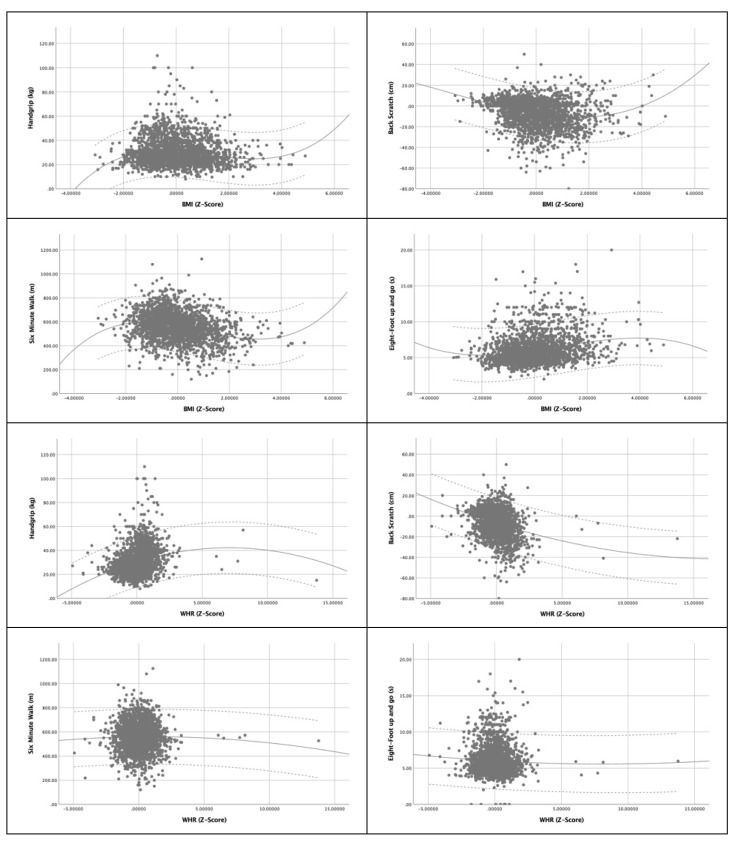
Lowess fit plots of Handgrip, Back Scratch, Six-Minute Walk, and Eight-Foot Up-and-Go vs. anthropometric indices (BMI, WHR, WHtR, BAI, CI).

**Table 1 ijerph-21-00201-t001:** Characteristics of all participants.

	ALL n = 2687	Male n = 688	Female n = 1999	*p*
	Mean ± SD	Mean ± SD	Mean ± SD	
Age (years)	70.05 ± 5.51	71.16 ± 5.95	69.69 ± 5.31	*p* < 0.001
Age Interval (n, %)				
59–64	367, 13.70%	74, 10.7%	293, 14.6%	-
65–69	1008, 37.50%	237, 34.5%	771, 38.5%	-
70–74	795, 29.60%	215, 31.3%	580, 29.0%	-
75–79	367, 13.7%	96, 14.0%	271, 13.5%	-
80–90	150, 5.6%	66, 9.4%	84, 4.3%	-
Physical Fitness				
Handgrip (kg)	30.16 ± 11.27	41.63 ± 12.70	26.29 ± 7.48	*p* < 0.001
Arm Curl (n)	21.90 ± 5.85	21.02 ± 5.26	22.19 ± 6.01	*p* < 0.005
Chair Stand Test (n)	17.24 ± 4.93	18.35 ± 4.92	16.86 ± 4.87	*p* < 0.001
Chair Sit and Reach (cm)	0.78 ± 8.64	−1.55 ± 9.70	1.56 ± 8.11	*p* < 0.001
Back Scratch (cm)	−6.74 ± 13.20	−12.07 ± 15.21	−4.91 ± 11.90	*p* < 0.001
Six-Minute Walk (m)	558.33 ± 115.72	607.09 ± 115.52	541.87 ± 111.09	*p* < 0.001
Two-Minute Step Test (n)	111.70 ± 46.50	116.9.7 ± 49.2	109.9 ± 45.50	*p* = 0.021
Eight-Foot Up-and-Go (s)	6.00 ± 1.97	5.46 ± 1.88	6.18 ± 1.97	*p* < 0.001
Anthropometric Indices				
Height (cm)	160.98 ± 8.47	169.71 ± 7.39	158.03 ± 6.56	*p* < 0.001
Weight (kg)	72.68 ± 12.51	79.68 ± 10.85	70.31 ± 12.15	*p* < 0.001
BMI (kg/m^2^)	28.03 ± 4.41	27.64 ± 3.37	28.16 ± 4.70	*p* = 0.014
Fat %	33.20 ± 8.13	25.60 ± 6.53	35.73 ± 6.95	*p* < 0.001
WC (cm)	93.92 ± 12.10	98.52 ± 9.83	92.35 ± 12.40	*p* < 0.001
HC (cm)	105.33 ± 10.92	102.94 ± 9.00	106.14 ± 11.38	*p* < 0.001
WHR	0.89 ± 0.09	0.96 ± 0.08	0.87 ± 0.35	*p* < 0.001
WHtR	0.58 ± 0.08	0.58 ± 0.06	0.58 ± 0.08	*p* = 0.250
BAI (%)	33.81 ± 6.79	28.68 ± 4.94	35.57 ± 6.44	*p* < 0.001
CI	1.29 ± 0.11	1.32 ± 0.08	1.27 ± 0.11	*p* < 0.001

Abbreviations: BAI, body adiposity index; BMI, body mass index; CI, conicity index; Fat %, body fat percentage; HC, hip circumference; WC, waist circumference; WHR, waist-to-hip ratio; WHtR, waist-to-height ratio.

**Table 2 ijerph-21-00201-t002:** Regression coefficients and their confidence intervals of anthropometric indices in association with Handgrip, Back Scratch, Six-Minute Walk, and Eight-Foot Up-and-Go in linear regression with adjustments for age and gender.

Anthropometric Indices (z-Score)	Handgrip			Back Scratch			Six-Minute Walk			Eight-Foot Up-and-Go		
	Coef	95%		Coef	95%		Coef	95%		Coef	95%	
BMI	−0.051	−0.103	−0.003	−0.083 *	−0.147	−0.013	−0.266 **	−0.319	−0.209	0.272 **	0.198	0.340
WHR	0.014	−0.039	0.086	−0.196 **	−0.256	−0.134	−0.080 *	−0.139	−0.020	0.042	−0.043	0.116
WHtR	−0.088 *	−0.139	−0.043	−0.129 **	−0.201	−0.053	−0.242 **	−0.306	−0.167	0.218 **	0.126	0.314
BAI	−0.125 *	−0.187	−0.070	−0.047	−0.112	0.029	−0.214 **	−0.278	−0.153	0.230 **	0.135	0.318
CI	−0.082 **	−0.134	−0.031	−0.093 **	−0.157	−0.023	−0.147 **	−0.223	−0.067	0.105	−0.019	0.220

** *p* < 0.001 * *p* < 0.05.

**Table 3 ijerph-21-00201-t003:** Mean and standard deviation of Handgrip, Back Scratch, Six-Minute Walk, and Eight-Foot Up-and-Go according to cutoff points of anthropometric indices as markers of obesity levels.

Anthropometric Indices	Handgrip (kg)	Back Scratch (cm)	Six-Minute Walk (m)	Eight-Foot Up-and-Go (s)
BMI (kg/m^2^)				
<27	30.98 ± 11.70 ^a^	−3.08 ± 11.35 ^a^**	591.15 ± 112.43 ^a^**	5.55 ± 1.61 ^a^**
≥27 a <30	30.62 ± 11.24 ^a^	−8.95 ± 13.25 ^b^	554.58 ± 110.51 ^b^**	6.14 ± 2.01 ^b^
≥30 a <35	28.99 ± 10.39 ^a^	−10.23 ± 13.61 ^b^	523.76 ± 110.40 ^c^**	6.42 ± 2.10 ^b^
≥35	26.05 ± 8.55 ^b^**	−8.15 ± 16.33 ^b^	480.54 ± 109.33 ^d^**	7.24 ± 2.42 ^c^**
WHR				
<0.90(male); <0.85 (female)	27.96 ± 9.20	−3.21 ± 10.85	564.49 ± 118.03	6.07 ± 2.02
≥0.90(male); ≥0.85 (female) [36]	31.25 ± 12.02 **	−8.46 ± 13.88 **	556.40 ± 114.68 *	5.96 ± 1.96
WHtR				
<0.60	31.17 ± 11.62	−3.70 ± 11.27	585.55 ± 115.41	5.67 ± 1.79
≥0.60 [38]	28.71 ± 10.62 **	−10.89 ± 14.44 **	521.88 ± 106.39 **	6.45 ± 2.14 **
BAI				
<28.0 (male); 36.0 (female)	31.05 ± 11.32	−3.47 ± 10.89	588.40 ± 114.78	5.56 ± 1.61
≥28.0 (male); 36.0 (female) [39]	29.08 ± 11.15 **	−10.53 ± 14.55 **	523.90 ± 107.52 **	6.52 ± 2.24 **
CI				
<1.275 (male); <1.285 (female)	28.92 ± 10.15	−3.96 ± 11.04	571.66 ± 116.19	5.88 ± 1.89
≥1.275 (male); >1.285 (female) [40]	31.20 ± 11.96 **	−9.02 ± 14.40 **	548.27 ± 115.24 **	6.13 ± 2.02 *

^a^ Categories homogeneous among themselves (column) and with significant differences with other categories (^b, c^); ^b^ Categories homogeneous among themselves (column) and with significant differences with other categories (^a, c^); ^c^ Categories homogeneous among themselves (column) and with significant differences with other categories (^a, b^); ** *p* < 0.001 * *p* < 0.05.

## Data Availability

The data presented in this study are available on request from the corresponding author.

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
