# Peer review of "Is There a Relationship between Anthropometric Indices and Muscular, Motor, and Cardiorespiratory Components of Health-Related Fitness in Active European Older Adults?"

_ijerph, 2024, doi:10.3390/ijerph21020201_

Round 1

Reviewer 1 Report

Comments and Suggestions for Authors

Dear authors.

First of all, my congratulations on your efforts. However, it is necessary to mention some aspects that can clarify the intentions of the work in order to make it more understandable to the possible reader.

- The work itself causes me some conflict, insofar as one of the components of healthy physical condition is, precisely, BMI. Therefore, it would be interesting if the authors could address this aspect, clarifying it in some way so as not to create doubts in the reader's mind.

- Abstract: Gender is mentioned as a determinant of physical fitness and anthropometric parameters. In what way? Moreover, this is not something to be considered in the planetary target... Why are anthropometric indices treated in a differentiated way, taking them out of healthy physical condition, when one of the tests included in the SFT is, precisely, the assessment of BMI?

- Introduction: (lines 48-49) Perhaps it should be mentioned that the loss of physical condition accentuates the detailed aspects. Even more so when, precisely in later lines, aspects such as the practice of physical activity or a sedentary lifestyle are mentioned as determining factors; (line 53) mention is made of the concept of "healthy physical condition", and I think it is more appropriate to refer to "helth-related fitness"; (line 53) there is a spelling error when it says ...is of vital....

- 2.1. Participants: It is necessary to detail the intervention on which the work is based.

- 2.2. Assessment: I think the anthropometric indices should be presented in the same way as the SFT evidence is presented, below; (line 118) I think this is an error, as it refers to BMI via BIA; (lines 117-122) In addition to those included here, the abstract refers to waist-to-hip ratio and others. In this case, it would be necessary to mention all of them. Also, obviously, in the results; I think it would be good to introduce a little bit, indicating something like flexibility is assessed by two tests.... This is extensible to all the capacities referred to in the document; BMI, which is part of the SFT is not presented here, although it is incorporated earlier. This is something that should be explained beforehand; (lines 202-217) Why is another measure of strength used?

- 3. Results: (lines 237-238) Is it to be understood that you want to refer to grassroots differences; (line 240 onwards) What does gender have to do with this? The SFT and certain anthropometric measures were used, applicable to both men and women, right?; Based on what criteria are the age ranges established; Sorry, I don't understand the results. An intervention is mentioned which, now, does not appear in any case; Is the effect of the intervention not evaluated?

- 4. Discussion: (line 335) Should low levels of muscle strength be understood; (line 351) Perhaps, instead of mentioning the exercise that makes up the test used, it would be more interesting to refer to the fact that it is about upper body strength; (line 358) The method mentions participation in the In Common Sport Plus Project in the years 2021-2022-2023. It makes no sense to make use of a programme, which, by the way, is not described in any way, and then indicate that it has not been taken into account. What is the purpose of the intervention then? This needs to be clarified; (line 358) what has diet got to do with it? This would, in any case, be another possible variable to consider.

- 5. Conclusions: (lines 368 - 370) But this was not the objective, was it? The objective was focused on the relationship between HRF and anthropometric measures.

Best regards.

Comments on the Quality of English Language

Without being an expert, I believe that there are expressions that could be improved for the better understanding of the readers.

Author Response

RESPONSE TO REVIEWERS

Manuscript Number: ijerph-2785625 

Title: “ Is there a relationship between anthropometric indices and the physical Fitness in European active older adults?”

The authors would like to thank the editor and the  reviewer 1 for the analysis of their work as well as for the considerations made and proposals for improvement. Based on them, the authors have made the following contributions to the document. Everything we have changed has been highlighted in green.

Revisor 1.

Dear authors. First of all, my congratulations on your efforts. However, it is necessary to mention some aspects that can clarify the intentions of the work in order to make it more understandable to the possible reader.

Q1.1. The work itself causes me some conflict, insofar as one of the components of healthy physical condition is, precisely, BMI. Therefore, it would be interesting if the authors could address this aspect, clarifying it in some way so as not to create doubts in the reader’s mind.

A1.1. The authors know that body composition (BMI) is a defining parameter of healthy physical condition, and this is identified in the paper (lines 60 to 63). The authors are also aware of the existence of numerous anthropometric indices, so the purpose of the study is to identify the degree of relationship between these indices and the muscular, motor and cardiorespiratory components of health-related fitness. Based on the reviewer’s comments, the authors have redrafted the introductory section and adapted the title and objective of the article.

Q1.2. Abstract: Gender is mentioned as a determinant of physical fitness and anthropometric parameters. In what way? Moreover, this is not something to be considered in the planetary target... Why are anthropometric indices treated in a differentiated way, taking them out of healthy physical condition, when one of the tests included in the SFT is, precisely, the assessment of BMI?

A1.2. The authors appreciate the reviewer’s comments. The authors, when referring to gender,would like to point out that the values ​​of the component of health-related fitness are not comparable between  men and women, as there are significant differences (see table 1). Therefore, the entire analysis was carried out adjusting for gender. The authors aim to identify which anthropometric index has the highest degree of association with the muscular, motor and cardiorespiratory components.

Q1.3 Introduction: (lines 48-49) Perhaps it should be mentioned that the loss of physical condition accentuates the detailed aspects. Even more so when, precisely in later lines, aspects such as the practice of physical activity or a sedentary lifestyle are mentioned as determining factors; (line 53) mention is made of the concept of “healthy physical condition”, and I think it is more appropriate to refer to “health-related fitness”; (line 53) there is a spelling error when it says ...is of vital....

A1.3. The authors thank the reviewer for their comments and have incorporated his input into the introduction by rewriting it.

Q1.4.  Participants: It is necessary to detail the intervention on which the work is based.

A1.4. The authors have indicated in the participants section, the criteriaused when defining active older adults.

Q1.5. Assessment: I think the anthropometric indices should be presented in the same way as the SFT evidence is presented, below; (line 118) I think this is an error, as it refers to BMI via BIA; (lines 117-122) In addition to those included here, the abstract refers to waist-to-hip ratio and others. In this case, it would be necessary to mention all of them. Also, obviously, in the results; I think it would be good to introduce a little bit, indicating something like flexibility is assessed by two tests.... This is extensible to all the capacities referred to in the document; BMI, which is part of the SFT is not presented here, although it is incorporated earlier. This is something that should be explained beforehand; (lines 202-217) Why is another measure of strength used?

A1.5. The authors agree with the reviewer’s comments and have rewritten the Evaluation section incorporating the information and clarifications requested by the reviewer.

Q1.6 Results: (lines 237-238) Is it to be understood that you want to refer to grassroots differences; (line 240 onwards) What does gender have to do with this? The SFT and certain anthropometric measures were used, applicable to both men and women, right?; Based on what criteria are the age ranges established; Sorry, I don’t understand the results. An intervention is mentioned which, now, does not appear in any case; Is the effect of the intervention not evaluated?

A1.6. The authors would like to apologize to the reviewer, as the translation process has caused a lot of confusion in this section. The authors have revised and rewritten the it.

Q1.5. Discussion: (line 335) Should low levels of muscle strength be understood; (line 351) Perhaps, instead of mentioning the exercise that makes up the test used, it would be more interesting to refer to the fact that it is about upper body strength; (line 358) The method mentions participation in the In Common Sport Plus Project in the years 2021-2022-2023. It makes no sense to make use of a programme, which, by the way, is not described in any way, and then indicate that it has not been taken into account. What is the purpose of the intervention then? This needs to be clarified; (line 358) what has diet got to do with it? This would, in any case, be another possible variable to consider.

A1.5. The authors have taken into account the reviewer’s comments and have rewritten that paragraph.

Q.1.6.  Conclusions: (lines 368 - 370) But this was not the objective, was it? The objective was focused on the relationship between HRF and anthropometric measures.

A1.6. The authors fully agree with the reviewer’s comments and have rewritten this section taking the comments into account.

Q1.7. Without being an expert, I believe that there are expressions that could be improved for the better understanding of the readers.

A1.7. The documents have been reviewed by a sworn translator and expert on this  field.

Reviewer 2 Report

Comments and Suggestions for Authors

Dear authors,

The structure and development of the work is well defined and well thought out. The theoretical foundation is relevant and the methodological development is appropriate for the type of study.

Before publication, in my opinion, a few minor revisions are needed, as listed below:

Line 253, Table 1. Age Interval (n, %), I suggest introducing the 59-64 interval instead of <65 and the 80-90 age interval instead of >80.

In the Materials and Methods part of the Participants section, I think it should also be specified in the paper, not only in the Acknowledgments, which European countries the participants come from, and if possible, the percentage by country and their gender. Authors must specify the exact period of time in which the databases were collected and the evaluation was carried out. At the same time, it must be specified how many participants were eliminated following the application of the exclusion criteria? It would be useful to introduce the flowchart of the selection process.

Kind regards,

Author Response

Manuscript Number: ijerph-2785625 

Title: “ Is there a relationship between anthropometric indices and the physical Fitness in European active older adults?”

The authors would like to thank the editor and the reviewer 2 for the analysis of their work as well as for the considerations made and proposals for improvement. Based on them, the authors have made the following contributions to the document. Everything we have changed has been highlighted in green.

Reviewer 2.

Dear authors. The structure and development of the work is well defined and well thought out. The theoretical foundation is relevant and the methodological development is appropriate for the type of study. Before publication, in my opinion, a few minor revisions are needed, as listed below:

Q2.1 Line 253, Table 1. Age Interval (n, %), I suggest introducing the 59-64 interval instead of <65 and the 80-90 age interval instead of >80.

A2.1. The authors have incorporated the requested change

Q2.2. In the Materials and Methods part of the Participants section, I think it should also be specified in the paper, not only in the Acknowledgments, which European countries the participants come from, and if possible, the percentage by country and their gender. Authors must specify the exact period of time in which the databases were collected and the evaluation was carried out. At the same time, it must be specified how many participants were eliminated following the application of the exclusion criteria? It would be useful to introduce the flowchart of the selection process.

A2.2. The authors consider the reviewer’s comments to be relevant and the requested information has been incorporated as well as the flowchart

Reviewer 3 Report

Comments and Suggestions for Authors

The topic addressed in the manuscript would interest the IJERPH readers. The purpose of the study is to examine the association between anthropometric indices and variables defining healthy physical condition in active European older adults. The results of the study found that there are correlations between specific anthropometric index, such as BMI and WHtR.

Major concerns

The rationale of the study needs to be stronger. Based on information provided in the manuscript, it seems studies in the past had examined various physical fitness levels and its associations with obesity among older adults. The results of the study did not offer much new information regarding the topic. Additionally, the results presented does not match with the purpose listed in the introduction.

Minor concerns

Abstract

- The purpose listed in the abstract does not match with the purpose listed in the introduction.

Introduction

- Need to define active aging in pg. 2 line 63. Without it, readers will not know what you mean by active aging.

- Be consistent with wording, either use older adults or the elderly.

- Pg. 2 line 77-79: overweight and obesity could be due to variety of factors as mentioned in the paper. Nutritional status are not the sole reasons why older adults are not physically fit.

- Pg. 2 line 76: need to provide definition on “definitive parameters” of healthy physical conditions with citations.

Materials and Methods

- Need to define “active” older adults.

- Need to provide details on the comparison tests (t-test and chi-square). It is unclear what groups were compared.

- It in unclear whether the regression included all z-scores of anthropometric indices or different regressions were used for each anthropometric indices.

- Need to provide details on the parameter of obesity levels for each anthropometric indices.

Results

- Be mindful about the correlation provided with linear regression also included other variables in the regression, if performed an adjusted linear regression.

- It will be a good idea to present the proportion of participants who were obese or not.

- In table 3, please follow the American English grammar rule of presenting decimal with “.” rather than “,”.

Discussion

- Rather than identifying previous studies found similar or dissimilar results, it is important to explain the results on the “why”, such as why does it make sense or not make sense for WHtR to have the strongest association with healthy physical condition.

- There is a need to identify the implication of the study.

Author Response

RESPONSE TO REVIEWERS

Manuscript Number: ijerph-2785625 

Title: “ Is there a relationship between anthropometric indices and the physical Fitness in European active older adults?”

The authors would like to thank the editor and the  reviewer 3 for the analysis of their work as well as for the considerations made and proposals for improvement. Based on them, the authors have made the following contributions to the document. Everything we have changed has been highlighted in green.

Reviewer 3

The topic addressed in the manuscript would interest the IJERPH readers. The purpose of the study is to examine the association between anthropometric indices and variables defining healthy physical condition in active European older adults. The results of the study found that there are correlations between specific anthropometric index, such as BMI and WHtR.

Q3.1. Major concerns. The rationale of the study needs to be stronger. Based on information provided in the manuscript, it seems studies in the past had examined various physical fitness levels and its associations with obesity among older adults. The results of the study did not offer much new information regarding the topic. Additionally, the results presented does not match with the purpose listed in the introduction.

A3.1. The paper has been reviewed and rewritten to respond to the research objectives.

Q3.2. Minor concerns. Abstract. The purpose listed in the abstract does not match with the purpose listed in the introduction.

A3.2.The abstract has been revised based on the reviewer’s contributions

Q3.3. Introduction. Need to define active aging in pg. 2 line 63. Without it, readers will not know what you mean by active aging.

A3.3. This concept (active aging) has been defined in the introductory section. In the participants section, a reference to the criteria for considering active older adults has been included.

Q3.4Be consistent with wording, either use older adults or the elderly.

A3.4. The authors agree with what was indicated by the reviewer and have proceeded to replace elderly with older adults.

Q3.5. Pg. 2 line 77-79: overweight and obesity could be due to variety of factors as mentioned in the paper. Nutritional status are not the sole reasons why older adults are not physically fit.

A3.5. The authors agree with the reviewer's instructions and have proceeded to rewrite these lines

Q3.6. Pg. 2 line 76: need to provide definition on “definitive parameters” of healthy physical conditions with citations. 

A3.6. The authors agree the reviewer's instructions and have proceeded to rewrite these lines.

Q3.7. Materials and Methods. Need to define “active” older adults.

A3.7. This concept has been defined in the participants section, indicating the criteria for considering  older adults active.

Q3.8. Need to provide details on the comparison tests (t-test and chi-square). It is unclear what groups were compared.

A3.8. This information has been incorporated into the statistical analysis section.

Q3.9. It in unclear whether the regression included all z-scores of anthropometric indices or different regressions were used for each anthropometric indices.

A3.9. In each regression analysis, only the anthropometric index under analysed and the muscular, motor or cardiorespiratory components of health-related fitness were included.

Q3.10. Need to provide details on the parameter of obesity levels for each anthropometric indices.

A3.10. The authors would like to indicate that the information requested by the reviewer was included in Table 3 and reference was made to the author on which it is based.

Q3.11. Results. Be mindful about the correlation provided with linear regression also included other variables in the regression, if performed an adjusted linear regression.

A3.11. The authors are aware of the implication of other variables when performing the adjusted linear regression analysis, but this has been done this way because it is reality, since the variables are not presented in isolation from each other.

Q3.12. It will be a good idea to present the proportion of participants who were obese or not.

A3.12. The authors would like to point out that this information was not included, since we did not consider it related to the object of study, but the cut-off point of obesity was taken into account according to the anthropometric index studied (see table 3).

Q3.13. In table 3, please follow the American English grammar rule of presenting decimal with “.” rather than “,”.

A3.13. The authors have made the adaptation suggested by the reviewer in table 3.

Q.3.14.Discussion. Rather than identifying previous studies found similar or dissimilar results, it is important to explain the results on the “why”, such as why does it make sense or not make sense for WHtR to have the strongest association with healthy physical condition.

Q3.14. The authors have incorporated, in the discussion, the approach that the reviewer suggested to them.

Q.3.15.There is a need to identify the implication of the study.

A3.15. The authors, in the discussion section, have indicated the implications that the results obtained may have on the control of the health status of active older people.